# Neural correlations enable invariant coding and perception of natural stimuli in weakly electric fish

Michael G Metzen, Volker Hofmann, Maurice J Chacron*

Department of Physiology, McGill University, Montreal, Canada

**Abstract** Neural representations of behaviorally relevant stimulus features displaying invariance with respect to different contexts are essential for perception. However, the mechanisms mediating their emergence and subsequent refinement remain poorly understood in general. Here, we demonstrate that correlated neural activity allows for the emergence of an invariant representation of natural communication stimuli that is further refined across successive stages of processing in the weakly electric fish *Apteronotus leptorhynchus*. Importantly, different patterns of input resulting from the same natural communication stimulus occurring in different contexts all gave rise to similar behavioral responses. Our results thus reveal how a generic neural circuit performs an elegant computation that mediates the emergence and refinement of an invariant neural representation of natural stimuli that most likely constitutes a neural correlate of perception.

## Introduction

Understanding how the brain processes sensory input in order to generate behavioral responses remains a central problem in neuroscience. Several studies have shown that neurons located in more central brain areas tend to respond more selectively to stimuli than neurons located in more peripheral brain areas (*Rolls and Tovee, 1995*; *Perez-Orive et al., 2002*; *Olshausen and Field, 2004*; *Hromadka et al., 2008*). At the same time, however, neural representations in more central brain areas become more robust to the differential patterns of sensory input associated with a given stimulus (e.g. those experienced when looking at the same object under different levels of illumination). Such invariant representations are thought to mediate context-independent object recognition and have been observed across sensory modalities (auditory: [*Bendor and Wang, 2005*; *Barbour, 2011*; *Bizley and Cohen, 2013*; *Rabinowitz et al., 2013*]; visual: [*Quiroga et al., 2005*; *Zoccolan et al., 2007*; *Rust and Dicarlo, 2010*; *DiCarlo et al., 2012*; *Sharpee et al., 2013*]; somatosensory: [*DiCarlo and Johnson, 1999*; *Pei et al., 2010*]; olfactory: [*Stopfer et al., 2003*; *Cleland et al., 2007*]). Modeling studies have proposed that such representations are built and refined in a feedforward manner across successive brain areas by using OR-like operations (*Hubel and Wiesel, 1962*; *Riesenhuber and Poggio, 1999*; *Kouh and Poggio, 2008*). Indeed, these studies suggest that combining neural activities that are locally invariant to complementary subsets of stimulus waveforms associated with a given object should then give rise to a more globally invariant representation. However, the actual computations used by the brain that give rise to invariant neural representations remain poorly understood to this day.

Wave-type gymnotiform weakly electric fish offer an attractive model system for studying the emergence and refinement of feature invariant neural representations because of well-characterized anatomy and relatively simple natural sensory stimuli that can easily be mimicked in the laboratory (*Chacron et al., 2011*; *Marsat et al., 2012*; *Marquez et al., 2013*; *Stamper et al., 2013*; *Krahe and Maler, 2014*; *Clarke et al., 2015b*). These fish generate a quasi-sinusoidal electric organ

*For correspondence: maurice. chacron@mcgill.ca

Competing interests: The authors declare that no competing interests exist.

**eLife digest** We can effortlessly recognize an object – a car, for example – in many different contexts such as when seen from behind, under different lighting levels or even from different viewpoints. This phenomenon is known as perceptual invariance: objects are correctly recognized, despite variations in exactly what is seen (or otherwise sensed). However, it is still not clear how the brain processes perceptual information to recognize the same object under a wide variety of contexts.

Some fish, such as the brown ghost knifefish, produce a weak electric signal that they can alter to communicate with other members of their species. A communication call may be produced in a variety of contexts that alter which aspects of the signal nearby fish detect. Despite this, fish tend to respond to a given communication call in the same way regardless of its context; this suggests that these fish also have perceptual invariance.

The communication calls of weakly electric fish can be easily mimicked in a laboratory and produce reliable behavioral responses, which makes these fish a good model for understanding how perceptual invariance might be coded in the brain. Therefore, Metzen et al. recorded the activity of the receptor neurons that first respond to communication calls in weakly electric fish. The results revealed that a given communication signal made the firing patterns of all receptor neurons in the fish's brain more similar to each other, regardless of the signal's context. This occurs despite the changes in context causing single receptor neurons to respond in different ways. At each stage of the process by which information is transmitted from the receptor neurons to neurons deeper in the brain, the similarity in the neurons' firing patterns is refined, thereby giving rise to perceptual invariance.

While perceptual invariance to a given object in different contexts is desirable, it is also important to be able to distinguish between different objects. This implies that neurons should respond similarly to stimuli associated with the same object and differently to stimuli associated with different objects. Further studies are now needed to confirm whether this is the case.

discharge (EOD) at a typical individual frequency, resulting in an electric field that surrounds the body. EOD perturbations are sensed by an array of electroreceptors that synapse onto pyramidal neurons within the hindbrain electrosensory lateral line lobe (ELL) that in turn synapse onto neurons within the midbrain Torus semicircularis (TS). When two conspecifics come into contact (<1 m), each fish experiences a sinusoidal amplitude modulation (i.e. beat) with a frequency that is equal to the difference between the two EOD frequencies. Natural communication signals or chirps consist of transient increases in EOD frequency and always occur simultaneously with the beat under natural conditions. Stimulus waveforms associated with chirps can be uniquely characterized by the increase in EOD frequency, duration, and the beat phase at onset. It has long been recognized that chirp stimulus waveforms are highly heterogeneous (*Zupanc and Maler, 1993*; *Hupé and lewis., 2008*). Interestingly, a recent study has shown that, while duration and frequency increase of chirps were both narrowly distributed, the beat phase at chirp onset was instead uniformly distributed over the entire beat cycle (*Aumentado-Armstrong et al., 2015*). This suggests that the observed heterogeneities in stimulus waveforms are, at least in part, caused by the fact that a chirp with given frequency increase and duration can occur with equal probability at any phase of the beat cycle.

Previous studies have shown that weakly electric fish perceive chirps as evidenced from behavioral responses (*Hupé and Lewis, 2008*). Electrophysiological studies have also characterized how chirps are encoded across successive stages of processing and notably, both afferents and ELL pyramidal neurons tend to respond differentially to different chirp stimulus waveforms (*Benda et al., 2005*; *2006*; *Marsat et al., 2009*; *Marsat and Maler, 2010*). TS neurons display large heterogeneities in their responses: while some respond in a manner that is similar to that of ELL neurons, others instead respond selectively to chirps through reliable and precisely timed action potentials (*Vonderschen and Chacron, 2011*). A recent study has, furthermore, shown that some TS neurons can respond similarly to different chirp stimulus waveforms (*Aumentado-Armstrong et al., 2015*). However, the critical questions of whether weakly electric fish can perceive that different waveforms

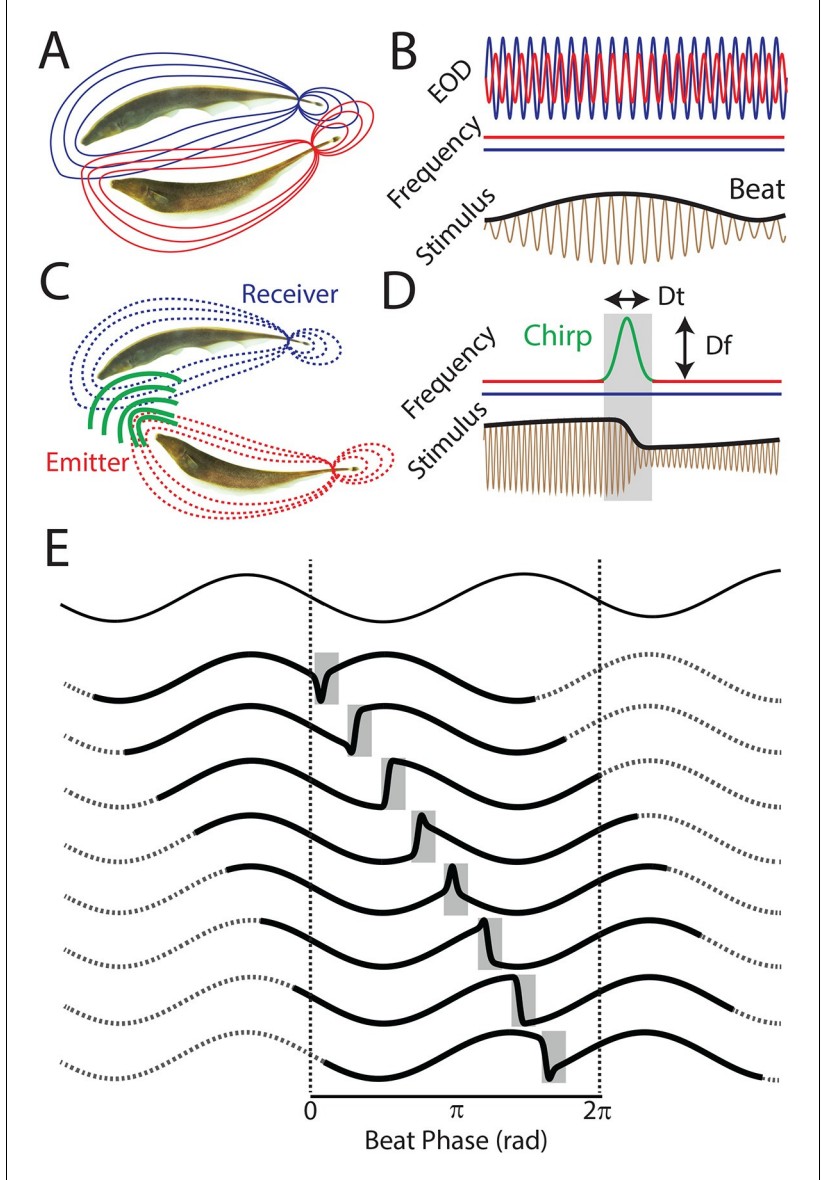

**Figure 1.** The same chirp stimulus gives rise to heterogeneous waveforms depending on time of occurrence within the beat cycle. (**A**) Two weakly electric fish with their electric organ discharges (EODs) in red and blue. (**B**) The EOD waveforms of both fish (top red and blue traces) show alternating regions of constructive and destructive interference when the instantaneous EOD frequencies do not vary in time (middle red and blue traces). Interference between the EODs leads to a sinusoidal amplitude modulation (i.e. a beat, bottom black trace) of the summed signal (bottom brown trace). (**C**) Schematic showing communication between the emitter (red) and receiver (blue) fish. (**D**) During a communication call, the emitter fish's EOD frequency (top red trace) transiently increased by maximum of $\Delta f$ for a duration $\Delta t$ (top green trace), while the receiver fish's EOD frequency (top blue trace) remains constant. The communication call results in a phase reset of the beat (bottom black trace). (**E**) Ongoing unperturbed beat (top) and stimulus waveforms (bold black traces within gray bands) resulting when a chirp with the same frequency increase $\Delta f$ and duration $\Delta t$ occurs at different phases during the beat cycle. We note that these waveforms are very similar to those recorded from actual fish (**Aumentado-Armstrong et al., 2015**).

The following source data and figure supplements are available for figure 1:

**Figure supplement 1.** EOD amplitude modulations resulting from chirps display large heterogeneities.

**Figure supplement 1—source data 1.** Source data for *Figure 1—figure supplement 1*.

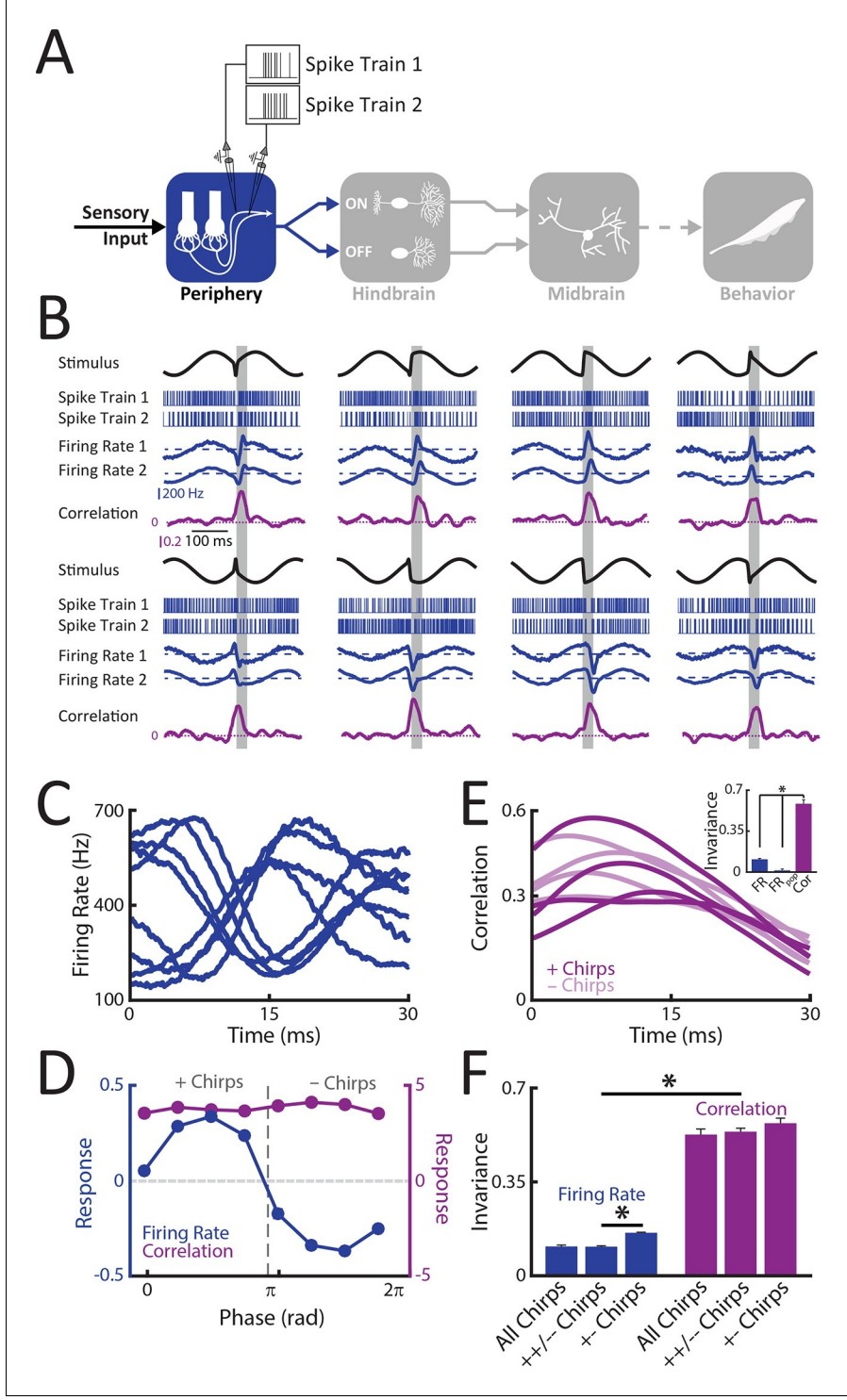

**Figure 2.** Correlated but not single peripheral afferent activity provides a representation that is invariant to the different patterns of stimulation resulting from the same chirp occurring at different phases within the beat cycle. (**A**) Schematic showing the different stages of processing in the electrosensory system. Recordings were made from single (N = 18) as well as pairs (N = 8) of afferents. (**B**) Example spike trains (blue, top), firing rates averaged across trials (middle, blue), and correlation coefficient (purple) from an example afferent pair to different stimulus waveforms associated with the same electrosensory object. Note that each stimulus was presented at least 20 times. The horizontal bars (shaded gray) represent the chirp window (30 ms) used for evaluation. (**C**) Population-averaged firing rate responses to the different stimulus waveforms during the 30 ms chirp window shown in B. (**D**)

*Figure 2 continued on next page*

*Figure 2 continued*

Population-averaged tuning curves obtained from single neuron firing rates (blue) and from the correlation coefficient (purple). '+ chirps' and '- chirps' were defined as the stimulus waveforms that gave rise to positive (i.e. increases in firing rate) and negative (i.e. decreases in firing rate) responses in afferents, respectively. (E) Correlation coefficient as a function of time in response to all stimulus waveforms. Inset: Invariance score computed from single afferents (left), from the population-averaged activity (center), and from the correlation coefficient (right). (F) Invariance score computed from single afferent activity (blue) and from correlated activity (purple) computed for all stimulus waveforms, '++/- - chirps', and '+- chirps'. '*' indicates statistical significance at the p = 0.05 level using a one-way ANOVA with Bonferroni correction.

The following source data and figure supplements are available for figure 2:

**Source data 1.** Source data for *Figure 2*.

**Figure supplement 1.** Phase invariant coding by single and correlated activity.

**Figure supplement 1—source data 1.** Source data for *Figure 2—figure supplement 1*.

might constitute signatures of the same chirp stimulus and, if so, the computations leading to the emergence and refinement of an invariant neural representation have not been investigated to date.

## Results

We studied the neural encoding and perception of natural communication signals by using a systems level approach in which we recorded from peripheral receptor afferents, ELL pyramidal neurons, TS neurons, and ultimately behavioral responses at the organismal level. In the absence of communication signals (i.e. when both EOD frequencies are constant), interference between the emitter and receiver fish's EODs gives rise to a sinusoidal amplitude modulation as mentioned above (i.e. the beat) (*Figure 1A,B*). Natural communication stimuli (i.e. chirps) occur during social interactions in which the emitter fish sends the signal to the receiver fish (*Figure 1C*). The chirp consists of one fish increasing its EOD frequency for a short duration (*Figure 1D*, top red/green trace), which, when considering the resulting stimulus sensed by the receiver fish, consists of a transient perturbation of the underlying beat (*Figure 1D*, black bottom trace). Because a chirp with a given EOD frequency time-course (i.e. 'identity') can occur with uniform probability at all phases of the beat (*Aumentado-Armstrong et al., 2015*), the resulting stimulus waveforms display significant heterogeneities (*Figure 1E*, gray bands, *Figure 1—figure supplement 1*). We therefore investigated whether and, if so, how neural responses across successive stages of processing to a chirp with given identity were similar irrespective of its occurrence within the beat cycle and, importantly, whether these constitute a neural correlate of behavior. We will henceforth refer to this concept as 'phase invariance'.

### Peripheral single receptor afferents display minimal phase invariance

We first recorded from single peripheral receptor afferents (*Figure 2A*) in response to different patterns of sensory input resulting from the same chirp stimulus occurring at different phases during the beat cycle (*Figure 1E*). Single afferent responses consisted of patterns of excitation and inhibition that strongly depended on the stimulus waveform that was presented (*Figure 2B*, blue curves). Indeed, population-averaged responses strongly varied depending on where the chirp occurred within the beat cycle (*Figure 2C*). Afferents were excited when the chirp occurred at beat phases lower than 180° and inhibited when it occurred at beat phases greater or equal to 180° (*Figure 2D*, blue curve). We shall henceforth refer to the stimulus waveforms that gave rise to excitation and inhibition as '+ chirps' and '- chirps', respectively. The similarity between single afferent firing rate responses to different waveforms during a 30 ms time window commensurate with chirp duration was then compared to that obtained between the stimulus waveforms during the same time window in order to measure phase invariance (see Methods). An invariance score near zero indicates that the response waveforms display heterogeneities that are similar to those displayed by the stimulus waveforms while a value near one implies that the response waveforms are all the same. We found

that the invariance score for single afferents was near zero (mean: 0.11 ± 0.01; max: 0.15; min: 0.07; population: 0.08) (*Figure 2E*, inset): this is because the distance between responses was always more or less equal to the distance between the corresponding stimulus waveforms (*Figure 2—figure supplement 1*). We note that this low value was not due to variability in the response of single afferents, as the population responses obtained by linearly summating the individual neural responses also displayed invariance scores near zero (*Figure 2E*, inset). We further note that computing invariance scores using only responses to subsets of chirps (i.e. those computed using only same type '+ +/- -' or opposite type '+-' chirp combinations) resulted in low values as well (*Figure 2F*, blue). Thus, we conclude that single peripheral afferent responses were minimally phase invariant as these neurons faithfully encoded the different patterns of sensory input resulting from the same chirp occurring at different phases within the beat cycle.

## Correlated activity between receptor afferents is phase invariant

We next investigated phase invariance for multiple afferents. To do so, we performed simultaneous recordings from afferent pairs, as well as recombined recorded activities from single afferents, in response to the patterns of sensory input resulting from the same chirp occurring at different phases within the beat cycle used above for single afferents. Similar results were seen when considering simultaneously recorded or recombined pairs: afferent activities were always more similar after chirp onset as afferent spiking activities were more synchronized either through excitation or inhibition (*Figure 2B*, blue curves compare Firing Rate 1 to Firing Rate 2). We thus quantified similarity by computing a time varying correlation coefficient between spike trains at short timescales (see Methods). Our results show that correlation transiently increased following chirp onset (*Figure 2B*, purple curves). Importantly, the time course of the correlation coefficient was similar for all stimulus waveforms (*Figure 2E*), as quantified by a large invariance score (mean: 0.53 ± 0.02; max: 0.67; min: 0.35; population: 0.61) (*Figure 2E*, inset & *Figure 2F*) that was not significantly different for either simultaneously recorded (0.51 ± 0.02) or recombined (0.53 ± 0.03) afferent activities (t-test, p=0.4). These high invariance scores were obtained because the distance between the time-varying correlation coefficient was always less than that between the corresponding stimulus waveforms (*Figure 2—figure supplement 1*). Similar results were observed when systematically varying the time scale at which correlations were computed up to ~60 ms (*Figure 2—figure supplement 1*). We conclude that correlated activity at short timescales in receptor afferents displays phase invariance.

## ON and OFF-type hindbrain ELL pyramidal neurons display higher phase invariance than receptor afferents

Information transmitted by neural activity is only useful to the organism if it is actually decoded downstream. To investigate whether the electrosensory system actually uses the fact that correlated afferent activity provides a phase invariant representation of natural communication signals, we recorded from hindbrain ELL pyramidal neurons (*Figure 3A*). These receive synaptic input from afferents and furthermore constitute the sole output neurons of the ELL (*Maler, 1979*; *Maler et al., 1981*). Pyramidal neurons can be classified as either ON or OFF-type based on whether they respond to increases in stimulus amplitude through increases or decreases in firing rate, respectively (*Saunders and Bastian, 1984*) (see [*Clarke et al., 2015b*] for review). Previous results have shown that pyramidal neurons respond to correlated afferent activity (*Berman and Maler, 1999*; *Middleton et al., 2009*). Thus, our working hypothesis is that, if single ON and OFF-type ELL pyramidal neurons actually decode increases in correlated afferent activity in response to natural communication stimuli, then they should: 1) display significantly more phase invariance than single afferents and 2) respond in a complementary fashion to different waveform subsets. Specifically, ON and OFF-type pyramidal neurons should respond more similarly with excitation and inhibition to the stimulus waveforms giving rise to synchronized excitation (i.e. '+ chirps') and inhibition (i.e. '- chirps') in afferents, respectively.

Our results were largely consistent with these predictions as '+ chirps' mostly gave rise to similar excitatory activity patterns in ON-type cells (*Figure 3B*, top rows, green; *Figure 3C*, dark green) and inhibitory activity patterns in OFF-type cells (*Figure 3B*, top rows, magenta; *Figure 3D*, light magenta). In contrast, '- chirps' mostly gave rise to patterns of inhibition in ON-type cells (*Figure 3B*, lower rows, green; *Figure 3C*, light green) and excitation in OFF-type cells (*Figure 3B*,

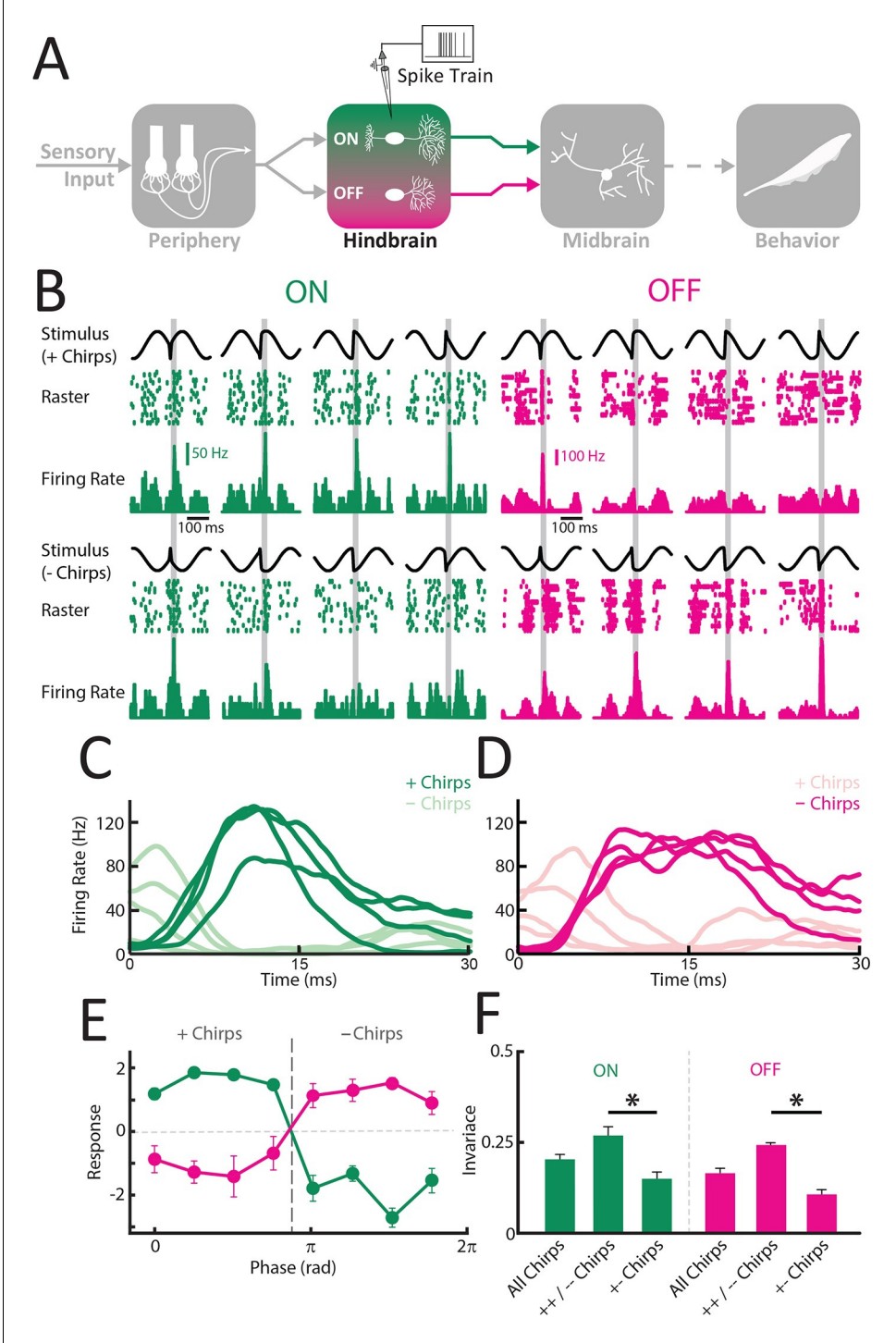

**Figure 3.** ON and OFF-type hindbrain ELL pyramidal neurons display more phase invariance by decoding correlated activity from peripheral afferents. (**A**) Schematic showing the different stages of processing in the electrosensory system. Recordings were made from ELL ON- (N = 22) and OFF-type (N = 9) pyramidal neurons. (**B**) Example responses of an ON-type (green) and an OFF-type neuron (magenta) to stimulus waveforms associated with the same electrosensory object. Shown are the stimulus waveforms, raster plots showing responses to 20 presentations of each stimulus, and trial-averaged time varying firing rates. Stimulus waveforms corresponding to '+ chirps' are shown above while those corresponding to '- chirps' are shown below. Note that excitatory responses of the ON-type neuron were mainly observed during + chirps while excitatory responses of the OFF-type neuron were found for - chirps. (**C,D**) Averaged time dependent firing rates for ON (**C**) and OFF (**D**) –type

*Figure 3 continued on next page*

*Figure 3 continued*

neurons in response to '+ chirps' (dark green) and '- chirps' (light green). (E) Tuning curves averaged from individual responses for ON- (green) and OFF (magenta) -type neurons. (F) Phase invariance scores computed from individual ON and OFF-type pyramidal neurons for all stimulus waveforms, same-type chirp combinations only ('++/- -'), and opposite-type chirp combinations only '+- chirps'. '*' indicates statistical significance at the p = 0.05 level using a one-way ANOVA.

The following source data and figure supplements are available for figure 3:

**Source data 1.** Source data for *Figure 3*.
**Figure supplement 1.** Responses of ELL pyramidal ON and OFF type populations to chirp stimuli.
**Figure supplement 1—source data 1.** Source data for *Figure 3—figure supplement 1*.

lower rows, magenta; *Figure 3D*, dark magenta). The complementarity of responses was further-more reflected by the fact that the population responses of ON and OFF cells were negatively corre-lated during responses to all waveforms (*Figure 3—figure supplement 1*). Consequently, the average tuning curves of ON and OFF-type cells were mirror images of one another (*Figure 3E*). Consistent with our predictions, ELL pyramidal neurons were significantly more phase invariant than single afferents (ELL ON; mean: $0.19 \pm 0.01$; max: 0.32; min: 0.03; population: 0.19; ELL OFF; mean: $0.16 \pm 0.01$; max: 0.21; min: 0.07; population: 0.16) (ANOVA with Bonferroni correction, $p<0.01$) (*Figure 3F*). Further analysis revealed that this was because the distance between responses was, on average, lower than that between the corresponding stimulus waveforms (*Figure 3—figure supple-ment 1*).

Why are ELL pyramidal neurons more phase invariant than single afferents? We hypothesized that the increased phase invariance is primarily due to the fact that ON cells respond similarly with excita-tion to '+ chirps' and with inhibition to '- chirps' while OFF cells instead respond similarly with inhibi-tion to '+ chirps' and with excitation to '- chirps'. These opposite responses to different subsets of chirp stimulus waveforms should limit phase invariance as the distances between the evoked responses will then be larger. To test this hypothesis, we computed the invariance score only for same-type chirps combinations (i.e. '++/- -') as well as for opposite-type chirps combinations (i.e. '+-'). We found that invariance scores computed for same-type chirp combinations were significantly larger than those computed for opposite-type chirp combinations for both ON (ANOVA; $p<0.01$) and OFF-type (ANOVA; $p<0.018$) pyramidal cells (*Figure 3F*). Further, average invariance score val-ues computed for opposite-type chirp combinations in ON ($0.15 \pm 0.02$) and OFF-type ($0.11 \pm 0.01$) ELL pyramidal cells were similar to those obtained for single afferents ($0.16 \pm 0.002$) (compare *Figures 3F* and *2F*). These results thus confirm our hypothesis that the increased phase invariance of ELL pyramidal cells is due to the fact that they respond more similarly to '+ chirps' and '- chirps' than single afferents. However, phase invariance in ELL is limited by the fact that ON and OFF-type pyramidal cells both respond in opposite fashion to different subsets of chirp stimulus waveforms (i.e. '+ chirps' and '- chirps').

## Midbrain TS Neurons display higher phase invariance than ELL pyramidal neurons

So far, we have shown that correlated but not single afferent activity provided a phase invariant representation of natural communication stimuli. Recordings from ELL pyramidal neurons revealed that these responded primarily through excitation to one half of the input patterns resulting from the same chirp occurring at different phases within the beat cycle and through inhibition to the other half, which limits phase invariance. Perhaps the simplest way to increase phase invariance is to sum the activities of ON and OFF cells. These responses should then be more similar and thus increase phase invariance. To test this hypothesis, we recorded from midbrain TS neurons (*Figure 4A*). Ana-tomical studies have shown that neurons within the midbrain TS receive only direct excitatory synap-tic input from multiple ON and OFF-type ELL pyramidal neurons (*Carr and Maler, 1985*), as

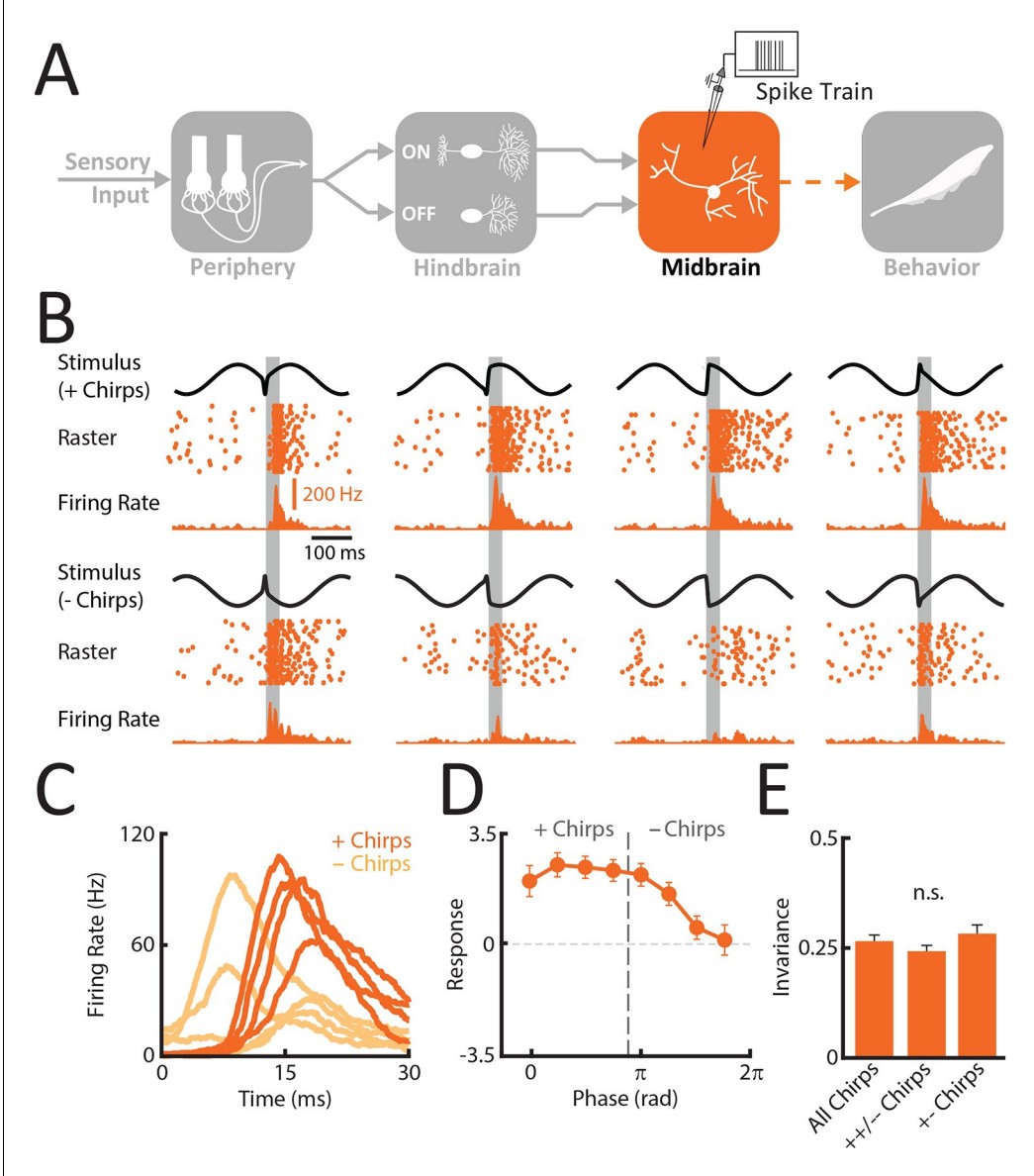

**Figure 4.** Midbrain TS neurons display invariant responses to the different patterns of stimulation resulting from the same chirp occurring at different phases within the beat cycle. (**A**) Schematic showing the different stages of processing in the electrosensory system. Recordings were made from TS neurons that receive balanced excitatory input from both ON and OFF-type ELL hindbrain neurons (N = 25). (**B**) Example responses of a TS neuron to stimulus waveforms associated with the same electrosensory object. Shown are the stimulus waveforms, raster plots showing responses to 20 presentations of each stimulus, and trial-averaged time varying firing rates. Stimulus waveforms corresponding to '+ chirps' are shown above while those corresponding to '- chirps' are shown below. (**C**) Averaged responses of TS neurons were similar when presented with stimulus waveforms associated with a given electrosensory object. (**D**) Tuning curve averaged from individual responses (mean ± SEM) for TS neurons. E. Invariance scores from single TS neurons for all stimulus waveforms, same-type chirp combinations only ('++/- -'), and opposite-type chirp combinations only '+-'.

The following source data and figure supplements are available for figure 4:

**Source data 1.** Source data for *Figure 4*.
**Figure supplement 1.** TS neurons respond more similarly and with excitation to all chirp waveforms.
**Figure supplement 1—source data 1.** Source data for *Figure 4—figure supplement 1*.

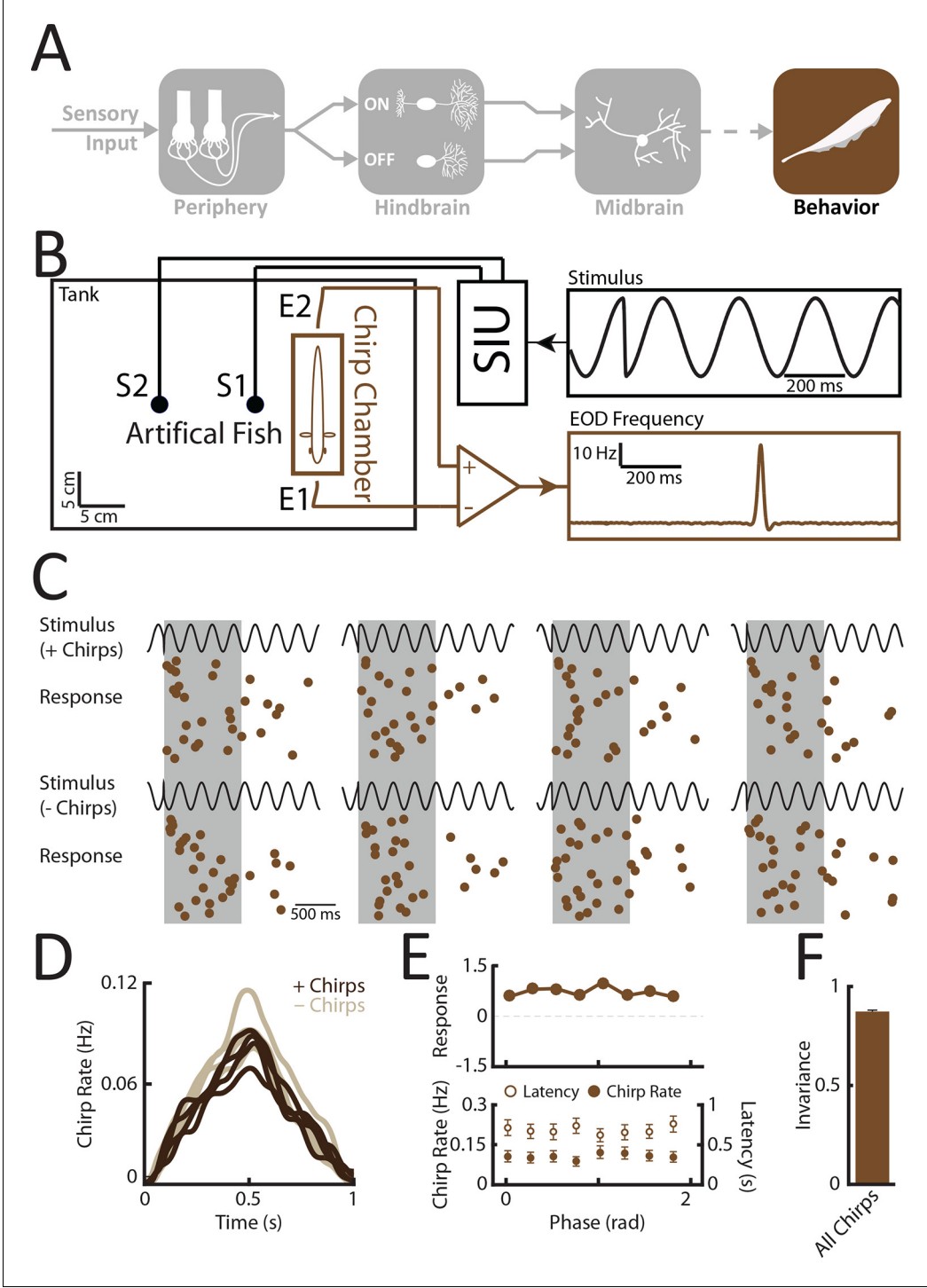

**Figure 5.** Behavioral responses indicate that perception is invariant to the different patterns of stimulation resulting from the same chirp occurring at different phases within the beat cycle. (**A**) Schematic showing the different stages of processing in the electrosensory system. We recorded behavioral responses from N = 29 fish. (**B**) Experimental setup. The fish was placed in an enclosure within a tank (chirp chamber). Stimuli were applied via two electrodes (S1 & S2) perpendicular to the fish's rostro-caudal axis. The fish's EOD frequency was recorded by a pair of electrodes positioned at the head and tail of the animal (E1 & E2). Behavioral responses consisted of communication stimuli characterized by transient increases in EOD frequency in response to the presented stimulus. (**C**) Raster plots showing behavioral responses (brown) to repeated presentations of different stimulus waveforms (black) associated with the same electrosensory object. (**D**) Population-averaged time dependent
*Figure 5 continued on next page*

*Figure 5 continued*

behavioral response rates in response to '+ chirps' (dark brown) and '- chirps' (light brown). (**E**) Population-averaged tuning curve (top), as well time-averaged behavioral response rates (bottom, filled circles) and latency to first chirp (bottom, open circles) as a function of beat phase. (**F**) Population-averaged invariance score computed from behavioral responses. EOD, electric organ discharge.

The following source data and figure supplements are available for figure 5:

**Source data 1.** Source data for *Figure 5*.
**Figure supplement 1.** Behavioral responses are phase invariant.
**Figure supplement 1—source data 1.** Source data for *Figure 5—figure supplement 1*.

reflected by their response properties (*McGillivray et al., 2012*). We thus hypothesized that single TS neurons should display significantly higher phase invariance than ELL neurons on average by responding primarily with excitation to most if not all patterns of stimulation input resulting from the same chirp occurring at different phases within the beat cycle.

Our results show that TS neurons indeed responded with excitation to most stimulus waveforms (*Figure 4B*) and that, consequently, responses were more similar than those of either single afferents or ELL pyramidal neurons (compare *Figure 4C* with *Figures 3C,D,* and *2C*). The response similarity was reflected in the population-averaged tuning curves (*Figure 4D*) as well as in the phase invariance score (mean: $0.26 \pm 0.02$; max: 0.49; min: 0.16; population: 0.4) that was significantly higher than that of ELL neurons (ANOVA with Bonferroni correction, p<0.01) when considering all waveforms (*Figure 4E*). This is because the distance between responses is considerably lower than the distance between the corresponding stimulus waveforms (*Figure 4—figure supplement 1*). Phase invariance score values obtained for the TS neuron population (0.4) furthermore approached those observed for correlated peripheral afferent activity (0.53). We note that the population-averaged response of TS neurons decreased for '- chirps' (*Figure 4D*): this is because some TS neurons responded similarly to ELL ON-type pyramidal cells and were thus inhibited by '- chirps'. However, TS neurons with larger phase invariance scores (N=13) were excited by all chirp waveforms as quantified by positive responses for all chirp waveforms (*Figure 4—figure supplement 1*). Further, computing phase invariance scores over either same type ('++/- -') or opposite type ('+-') chirp combinations led to values that were not significantly different from one another (++/- -: $0.24 \pm 0.02$; +-: $0.28 \pm 0.02$) (ANOVA; F = 1.3, p=0.28) and that were furthermore similar to those obtained using all waveforms ($0.26 \pm 0.02$). This confirms our hypothesis that TS neurons are, on average, more phase invariant than ELL pyramidal cells because a significant fraction ($\sim 50\%$) responds with excitation to both '+ chirps' and '- chirps'. The simplest explanation for this observation is that these neurons receive excitatory input from ELL ON and OFF-type pyramidal cells. This point is discussed further below.

## Weakly electric fish display phase invariant behavioral responses

Does the organism actually make use of the increased phase invariance observed when moving from peripheral to more central brain areas? To answer this important question, we recorded the animal's behavioral responses to patterns of stimulation resulting from the same chirp occurring at different phases within the beat cycle (*Figure 5A*) using a previously established behavioral paradigm (*Zupanc et al., 2006*; *Hupé et al., 2008*) (see thods) (*Figure 5B*). Specifically, we used the echo response in which an animal will reliably respond to a communication signal from a conspecific with a communication signal of its own (*Zupanc et al., 2006*). These occur during aggressive interactions between two conspecifics under natural conditions (*Hupé et al., 2008*). If our hypothesis is true, then we expect that all the presented waveforms will elicit similar behavioral responses. Confirming our hypothesis, the elicited behavioral responses were very similar (*Figure 5C,D*). Consequently, the response tuning was effectively independent of beat phase (*Figure 5E*, top). Importantly, other measures such as response latency and rate of occurrence also did not depend on beat phase (*Figure 5E*, bottom). Invariance values computed from behavioral responses were near unity (*Figure 5F*). This is because the distance between responses is considerably lower than the distance

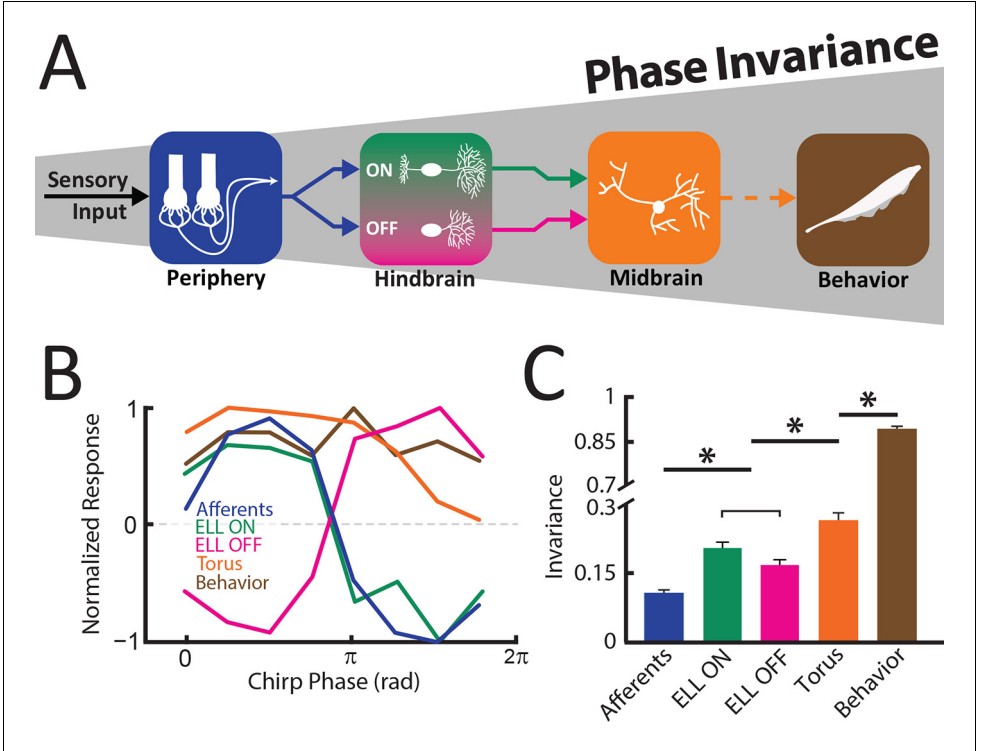

**Figure 6.** An invariant neural representation of natural communication stimuli emerges and is refined along ascending electrosensory brain areas, thereby giving rise to perception. (**A**) Schematic showing how invariance increases across successive stages of electrosensory processing, thereby leading to behavior. (**B**) Normalized response tuning curves obtained for each stage of sensory processing as well as behavior. Note that tuning curves become progressively more independent of beat phase. (**C**) Invariance progressively increases across successive stages of electrosensory processing as well as behavior downstream brain areas and behavior. '*' indicates statistical significance at the p = 0.01 level using a one-way ANOVA with Bonferroni correction.

The following source data is available for figure 6:

**Source data 1.** Source data for *Figure 6*.

between the corresponding stimulus waveforms (*Figure 5—figure supplement 1*). We conclude that the organism perceives the different patterns of stimulation input resulting from the same chirp occurring at different phases within the beat cycle in a similar fashion and thus displays perceptual phase invariance*Figure 6* .

## Summary

Taken together, our results provide the first evidence as well as an explanation as to how a phase invariant neural representation of natural communication stimuli first emerges in the hindbrain and is then refined in the midbrain in order to give rise to perception and behavior (*Figure 6A*). The tuning curves became progressively more independent of beat phase when comparing peripheral afferents, ON and OFF-type ELL pyramidal neurons, TS neurons, and behavior (*Figure 6B*), thereby leading to increased levels of phase invariance (*Figure 6C*). The implications of these results for electrosensory processing and for other systems are discussed below.

## Discussion

### Summary of results

We investigated the mechanisms by which neural representations of natural communication stimuli become progressively more phase invariant in the electrosensory system of weakly electric fish. We found that the responses of single peripheral afferents displayed minimal phase invariance, as these neurons faithfully encoded all the different waveforms resulting from the same chirp stimulus occurring at different phases within the beat cycle. In contrast, simultaneous recordings from afferent pairs revealed that these waveforms all gave rise to similar increases in correlated activity. ON and OFF-type ELL neurons displayed significantly higher phase invariance than single peripheral afferents. Indeed, both the ON and OFF-type subpopulations showed the greatest degree of phase invariance to waveforms giving rise to either excitation or inhibition in afferents. Responses from both subpopulations are most likely combined in the midbrain TS, thereby giving rise to a more phase invariant representation that, at the population level, approached that seen for correlated afferent activity. Finally, we showed that weakly electric fish displayed similar behavioral responses to stimulus waveforms generated by applying an identity-preserving transformation, which strongly suggests that the globally invariant representation observed in midbrain constitutes a neural correlate of perception. Our results thus reveal generic mechanisms that mediate the emergence and refinement of a neural representation that is invariant to the different patterns of stimulation input resulting from the same chirp occurring at different phases within the beat cycle.

### Coding of natural electro-communication stimuli

Previous studies have reported that the waveforms associated with natural communication stimuli were highly heterogeneous with respect to their time course (*Zupanc and Maler, 1993*) and that single peripheral afferents (*Benda et al., 2005*, *2006*) as well as ELL pyramidal neurons (*Marsat et al., 2009*; *Marsat and Maler, 2010*; *Vonderschen and Chacron, 2011*) display differential responses to these. While several studies have characterized the distributions of various natural communication stimulus attributes such as frequency increase, duration, or beat phase, for different beat frequencies (*Bastian et al., 2001*; *Zakon et al., 2002*; *Kolodziejski et al., 2005*; *Aumentado-Armstrong et al., 2015*), the tuning curves of electrosensory neurons to these have not been systematically investigated to date. Our results show that the different waveforms resulting from a chirp with given frequency increase and duration occurring at different phases within the beat cycle tend to give rise to more similar neural responses centrally. These most likely constitute a neural correlate of perception. Our electrophysiological and behavioral results thus support our initial hypothesis that the identity of a given chirp is determined by frequency increase and duration but not the phase of the beat at which it occurs at.

Our results further show that TS neurons displayed significantly less phase invariance than behavioral responses, indicating that further refinement in downstream brain areas is likely necessary. Such refinement is likely to occur in the nucleus electrosensorius, a diencephalic brain structure that receives direct input from TS (*Carr et al., 1981*), and likely involves integrating the activities of multiple TS neurons. The fact that the invariance score of the TS neuron population was on average higher than that of single neurons supports the latter statement. Importantly, we note here that the TS in weakly electric fish consists of 11 layers with roughly 50 cell types (*Carr and Maler, 1985*). Previous electrophysiological studies have revealed that TS neurons are highly heterogeneous and can be roughly classified in two groups: 'dense' neurons that respond to stimuli in a manner that is strongly reminiscent of that displayed by ELL pyramidal neurons; and 'sparse' neurons that respond selectively to a given class of natural electrosensory stimuli (*Vonderschen and Chacron, 2011*; *Sproule et al., 2015*). Within the sparse group, neurons can either respond selectively to movement (*Chacron et al., 2009*; *Vonderschen and Chacron, 2011*), increases in beat amplitude (i.e. envelopes) (*McGillivray et al., 2012*), or natural communication stimuli (*Vonderschen and Chacron, 2011*), indicating that parallel processing of behaviorally relevant stimulus features occurs in this area.

Interestingly, some TS neurons displayed phase invariance scores that were higher than those observed in single receptors or in ELL neurons. It is likely that these are members of the sparse group as they responded selectively to communication stimuli rather than the underlying beat. A previous study has shown that neurons with similar response profiles receive balanced excitatory

input from ON and OFF-type pyramidal cells (*Aumentado-Armstrong et al., 2015*) and, as such, most likely correspond to previously described 'ON-OFF' neurons that respond to both increases and decreases in the stimulus (*Partridge et al., 1981*; *Rose and Call, 1993*). We, however, note that these neurons correspond to a few if not a single cell type within TS (*Carr et al., 1981*; *Carr and Maler, 1985*), as partially reflected by the broad distribution of phase invariance scores observed in the current study in TS neurons. In fact, confirming previous results (*Vonderschen and Chacron, 2011*), a significant fraction of TS neurons displayed low-phase invariance scores comparable to those seen in ELL, indicating that they responded differentially (i.e. non-invariantly) to the different waveforms resulting from a given chirp occurring at different phases within the cycle of a beat. Interestingly, it was shown recently that both dense and sparse TS neurons project to higher brain areas (*Sproule et al., 2015*), indicating that these actually receive parallel sources of information including those transmitted by both phase invariant and non-phase invariant TS neurons. Phase invariant TS neurons would then give information that a chirp has occurred while non-phase invariant TS neurons would instead transmit contextual information about this chirp. While our behavioral results do not demonstrate whether the animal can actually perceive information transmitted by non-phase invariant TS neurons, it is likely that this information is behaviorally relevant.

However, further studies are needed to understand how TS neurons are tuned to other attributes of natural communication stimuli (i.e. frequency increase, duration, beat phase, beat frequency). This is important, as these have all been shown to carry behaviorally relevant information (*Bastian et al., 2001*; *Zakon et al., 2002*; *Kolodziejski et al., 2005*; *Hupé et al., 2008*). Thus, we predict that the organism will perceive communication signals whose attributes other than beat phase differ. We further hypothesize that, in more central brain areas, neural representations become more selective to chirp identity while at the same time becoming more tolerant to changes in waveform resulting from a given chirp occurring at different phases during the beat cycle. Specifically, we predict that peripheral afferents will faithfully encode the different waveforms associated with changes in a given feature irrespective of behavioral relevance. However, when moving more centrally from receptors to midbrain, neural responses should become more selective when varying chirp identity. If true, it will then be very interesting to understand the mechanisms by which neurons become selective to given stimulus attributes yet tolerant to others. While a previous study has shown that some TS neurons could respond differentially to communication signals with different frequency increases (*Vonderschen and Chacron, 2011*), the tuning properties of TS neurons to the behaviorally relevant features of natural communication stimuli have not been systematically investigated to date and should be the focus of future studies.

Finally, we note that we focused on small chirps or type II communication stimuli that tend to occur on top of low frequency beats (*Bastian et al., 2001*) because the associated waveforms displayed the largest heterogeneities (*Aumentado-Armstrong et al., 2015*). Recent studies have shown that type II communication stimuli can also occur on top of high frequency beats (*Walz, 2013*) and further studies are needed to understand whether the encoding strategies for these would differ from those described above. Also, big chirps or type I communication stimuli, as they mostly occur in high-frequency beat contexts, tend to display much less heterogeneity (*Bastian et al., 2001*; *Aumentado-Armstrong et al., 2015*). Previous studies have shown that the encoding strategies for type I and II communication stimuli strongly differ at the level of peripheral afferents (*Benda et al., 2005*, *2006*), ELL pyramidal cells (*Marsat et al., 2009*; *Marsat and Maler, 2010*; *Vonderschen and Chacron, 2011*), and it is likely that such differences are preserved in the midbrain (*Vonderschen and Chacron, 2011*). Further studies are needed to uncover how heterogeneities in type I communication stimuli are encoded in the electrosensory system. Also, we note that the large heterogeneities seen in stimulus waveforms associated with type II communication stimuli occurring at different beat phases might not be found in other weakly electric fish species. Indeed, the duration of type II communication stimuli in *Apteronotus albifrons* tend to be much larger than those of *Apteronotus leptorhynchus* considered in this study (*Kolodziejski et al., 2005*) and, as such, might not display the large degree of heterogeneity seen here.

## Implications for other systems

We have shown that the electrosensory system of weakly electric fish actually exploits the feature invariant representation that is given by correlated neural activity through ON and OFF-type ELL pyramidal neurons. We note that ON and OFF-type cells have been observed across species and

sensory modalities (*Wassle et al., 1981a*; *Wassle et al., 1981b*; *Wassle, 2004*; *Chalasani et al., 2007*; *Scholl et al., 2010*; *Gallio et al., 2011*; *Frank et al., 2015*) and, further, that downstream neurons receive excitatory inputs from both cell types (*Fairhall et al., 2006*; *Gollisch and Meister, 2008*). Thus, the discovered computations for generating and refining phase invariance uncovered in the electrosensory system that involve decoding correlated activity through ON and OFF-type neurons and their subsequent combination are also likely to be applicable. This is particularly the case in the primary visual cortex. Indeed, while simple cells respond selectively (*Movshon, 1975*), complex cells instead display responses that are invariant to the spatial phase of a drifting grating (*Movshon et al., 1978*). It is thought that this invariance arises, at least in part, through integration of input from multiple ON and OFF-type simple cells (*Hubel and Wiesel, 1962*; *Martinez and Alonso, 2003*). It is thus conceivable that correlated activity from simple cells could convey a spatial phase invariant representation that is then exploited by complex cells.

Further, correlations between the activities of neighboring neurons are observed ubiquitously in the brain (*Averbeck et al., 2006*) and are dynamically regulated by several factors such as attention (*Cohen and Maunsell, 2009*), behavioral state (*Vaadia et al., 1995*), and stimulus statistics (*Chacron and Bastian, 2008*; *Litwin-Kumar et al., 2012*; *Ponce-Alvarez et al., 2013*; *Simmonds and Chacron, 2015*). Previous studies have found that correlated but not single neuron activity can carry information about stimulus attributes (*deCharms and Merzenich, 1996*; *Ishikane et al., 2005*; *Metzen et al., 2015a*; *Metzen et al., 2015b*). Here, we have instead shown that correlated neural activity can mediate the emergence of a neural representation of behaviorally relevant stimuli that is phase invariant. We also note that, since previous studies have shown that noise correlations were negligible in electroreceptor afferents (*Chacron et al., 2005b*; *Metzen et al., 2015b*), our results focus on signal correlations that give rise to feature invariant representations of natural communication stimuli. Nevertheless, noise correlations are also dynamically regulated by stimulus attributes (*Chacron and Bastian, 2008*) and thus might provide information about sensory input. Future studies are needed to understand whether noise correlations can also transmit information about behaviorally relevant stimulus features.

Finally, we note that the natural communication stimuli considered here can be seen as high frequency transients that must be distinguished from a low frequency background (i.e. the beat). Thus, it is very likely that our results will be applicable to the problem of separating a fast moving object from a slow moving background as in the visual system (*Olveczky et al., 2003*) or to separating a high frequency transient sound occurring simultaneously with a low-frequency background in the auditory system (*Berti, 2013*).

## Conclusion

We investigated the neural coding strategies used by the electrosensory system to process natural communication stimuli of heterogeneous waveforms. Our results show that this system exploits the fact that correlated peripheral neuron activity provides an invariant representation through decoding synchronized excitation and inhibition by ON and OFF-type ELL pyramidal neurons, respectively. Midbrain neurons receiving input from both ON and OFF-type ELL pyramidal neurons displayed the most invariant responses and likely provide a neural correlate of the invariant perception seen at the organismal level.

# Materials and methods

All procedures were approved by McGill University's animal care committee under protocol number 5285.

## Animals

*Apteronotus leptorhynchus* specimens were acquired from tropical fish suppliers and acclimated to laboratory conditions according to published guidelines (*Hitschfeld et al., 2009*).

## Surgery and recordings

Surgical procedures have been described in detail previously (*Toporikova and Chacron, 2009*; *Vonderschen and Chacron, 2011*; *McGillivray et al., 2012*; *Deemyad et al., 2013*). Briefly, animals (N = 16) were injected with tubocurarine chloride hydrate (0.1 – 0.5 mg) before being transferred to an

experimental tank and respirated with a constant flow of water over their gills (~ 10 ml/min). A small craniotomy (~ 5 mm$^2$) was made above the hindbrain (for afferent and ELL pyramidal neuron recordings) or above the midbrain (for TS recordings), respectively. We used 3M KCl-filled glass micropipettes (30 MΩ resistance) to record from electroreceptor afferent axons as they enter the ELL (*Savard et al., 2011*; *Metzen and Chacron, 2015*; *Metzen et al., 2015b*) either simultaneously from pairs (N = 8) or from single units (N = 18). As similar results were obtained when considering pairs of simultaneously and non-simultaneously recorded afferents, data were pooled when computing correlated activity, these sample sizes were sufficient to access differences between responses to different stimulus waveforms based on previous studies (*Benda et al., 2005*; *Chacron et al., 2005b*; *Benda et al., 2006*). Importantly, it was previously shown that afferents do not display noise correlations (*Chacron et al., 2005b*; *Metzen et al., 2015b*). Extracellular recordings from ELL pyramidal (N = 31) and TS neurons (N = 25) were performed with metal-filled micropipettes (*Frank and Becker, 1964*; *Chacron et al., 2009*; *Chacron and Fortune, 2010*; *Simmonds and Chacron, 2015*). These sample sizes are similar to those used in previous studies. Baseline (i.e. in the absence of stimulation) firing rates for afferents, pyramidal cells, and TS neurons were 380 ± 72 Hz, 12.10 ± 1.74 Hz, and 3.35 ± 3.27 Hz. These are similar to previously reported values (*Chacron et al., 2005a*; *McGillivray et al., 2012*; *Clarke et al., 2015a*; *Metzen et al., 2015b*). We only recorded from neurons that responded to at least one chirp stimulus waveform. Recordings were digitized at 10 kHz (CED Power 1401 & Spike 2 software, Cambridge Electronic Design) and stored on a computer for subsequent analysis.

## Stimulation

The neurogenic electric organ of *A. leptorhynchus* is not affected by injection of curare-like drugs. Stimuli thus consisted of amplitude modulations of the animal's own EOD and were produced by first generating a sinusoidal waveform train with frequency slightly (20–30 Hz) greater than the EOD frequency that was triggered by the EOD zero crossing. This train is synchronized to the animal's EOD and will either increase or decrease EOD amplitude based on polarity and intensity. This train is then multiplied (MT3 multiplier, Tucker Davis Technologies) with an amplitude modulated waveform (i.e. the stimulus). The resultant signal is then isolated from ground (A395 linear stimulus isolator, World Precision Instruments) and delivered to the experimental tank via two chloritized silver wire electrodes located ~ 15 cm on each side of the animal (*Bastian et al., 2002*). Chirps consisting of Gaussian increases in frequency of 60 Hz and standard deviation of 14 ms, were generated on top of a beat with frequency $f_{beat}$ = 4 Hz and the resulting AM stimulus waveforms were used as done previously (*Vonderschen and Chacron, 2011*). Previous studies have shown that chirps occur with uniform probability over all phases of the beat (*Aumentado-Armstrong et al., 2015*). For experimental purposes, we systematically varied the beat phase at which the chirp occurred between 0 and 315 deg in increments of 45 deg, the corresponding AM waveforms (S1 to S8) are shown in *Figure 1E*. The same chirp stimuli were also used in the behavioral experiments.

## Analysis

All analyses were performed using custom-built routines in Matlab (The Mathworks, Natick, MA), these routines are freely available online (*Metzen et al., 2016*).

## Electrophysiology

Action potential times were defined as the times at which the signal crossed a suitably chosen threshold value. From the spike time sequence we created a binary sequence $R(t)$ with binwidth $\Delta t$ = 0.5 ms and set the content of each bin to equal the number of spikes which fell within that bin. PSTHs were obtained by averaging the neural responses across repeated presentations of a given stimulus with binwidth 0.1 ms and were smoothed with a 6 ms long boxcar filter.

## Correlation between the spiking activities of electrosensory afferents

We computed the cross-correlation coefficient between the spiking responses $R_i(t)$ and $R_j(t)$ of neurons $i$ and $j$ as was done previously (*Shea-Brown et al., 2008*; *Metzen et al., 2015b*). The time varying correlation coefficient was computed during a time window with duration 31.25 ms unless otherwise indicated that was translated in steps of 0.25 ms using:

$$\rho = \frac{|P_{R_iR_j}(0)|}{\sqrt{P_{R_iR_i}(0)P_{R_jR_j}(0)}} \tag{1}$$

Here, $P_{R_iR_i}$ is the cross-spectrum between $R_i(t)$ and $R_j(t)$, $P_{R_iR_i}(f)$ and $P_{R_iR_i}(f)$ are the power spectra of $R_i(t)$ and $R_j(t)$, respectively, and $|\ldots|$ denotes the absolute value. We note that our definition of the cross-correlation coefficient is equivalent to that used in previous studies (*Shadlen and Newsome, 1998*; *Chacron and Bastian, 2008*) and furthermore gives results similar to those obtained when computing the cross-correlation coefficient between spike counts during a time window of 40 ms (*de la Rocha et al., 2007*; *Metzen et al., 2015b*).

## Neural tuning

For each cell, the response to a given chirp stimulus waveform as:

$$\mathrm{Response} = \log\left(\frac{\mathrm{FR}_{\mathrm{chirp}}}{\mathrm{FR}_{\mathrm{beat}}}\right) \tag{2}$$

where $FR_{chirp}$ is the average firing rate over a time window of duration $T_{chirp}$ centered 15 ms after chirp onset and $FR_{beat}$ is the average firing rate during the beat. Note that response values can vary between $-\infty$ and $+\infty$. Negative response values imply inhibition and positive values imply excitation by the stimulus waveform, respectively. We used $T_{chirp}$ = 15 ms for electrosensory afferents and ELL pyramidal cells and $T_{chirp}$ = 20 ms for TS neurons. To obtain the tuning curve, the responses obtained for each chirp waveform were plotted as a function of beat phase for each cell and then averaged across the respective population.

## Quantifying neural response invariance

The phase invariance score was defined as:

$$Invariance = 1 - \frac{\sum_{i \neq j}\left[\frac{D(FR_i(t), FR_j(t)))}{D(S_i(t), S_j(t))}\right]}{(N_{chirps})(N_{chirps} - 1)} \tag{3}$$

where $N_{chirps}$ is the number of chirp stimulus waveforms, and $D(x,y)$ is a distance metric between x and y that was computed as (*Aumentado-Armstrong et al., 2015*):

$$D(x,y) = \frac{\sqrt{\left\langle (x - \langle x \rangle - y + \langle y \rangle)^2 \right\rangle}}{max\left[\frac{max(x)-min(x)}{\sqrt{2}}, \frac{max(y)-min(y)}{\sqrt{2}}\right]} \tag{4}$$

where $\langle \ldots \rangle$ denotes an average over a time window of 30 ms after chirp onset, $FR_i(t)$ is the PSTH response of a given cell to chirp stimulus waveform $S_i(t)$, and $max(\ldots)$, $min(\ldots)$ denote the maximum and minimum values, respectively. All responses were normalized prior to computing the distance metric. We note that, according to equation [3], the distance between responses to two different stimulus waveforms is actually normalized by the distance between the stimulus waveforms themselves. Thus, a neuron whose response faithfully encodes the detailed timecourse of the different chirp waveforms will not be considered invariant according to our definition. Invariance scores were computed for each individual cell and subsequently averaged across the respective populations. Some analyses were performed on TS neurons (N = 13) whose phase invariance score was greater than 0.25, which was chosen as the mean plus one standard deviation from our ELL data. We computed phase invariance for correlated activity as described above except that we used the time-course of the varying correlation coefficient as an input. Alternatively, to test that the low values obtained for receptor afferents were not due to variability, we also computed phase invariance scores for these by first averaging responses across the population.

## Behavior

To measure the chirp stimulus waveforms shown in *Figure 1—figure supplement 1*, fish were restrained by placing them in a 'chirp chamber' (*Deemyad et al., 2013*, *Metzen and Chacron, 2014*). Chirps were elicited by sinusoidal waveforms mimicking another fish's EOD whose frequency was set 10 Hz above the animal's own EOD frequency. The fish's EOD was recorded and the

instantaneous EOD frequency was computed from the inverse of the timing difference between successive zero crossings. Chirps were then identified as increases in the animal's own EOD frequency that exceeded 30 Hz (*Bastian et al., 2001*).

To measure the chirp echo response, each fish (N = 29) was restrained in a 'chirp chamber' as described previously (*Deemyad et al., 2013*, *Metzen and Chacron, 2014*). The EOD was measured between electrodes placed near the head and tail, amplified (model 1700 amplifier, A-M systems), digitized at 10 kHz sampling rate using CED 1401plus hardware and Spike2 software (Cambridge Electronic Design) (*Figure 5B*, E1 and E2), and stored on a computer hard disc for offline analysis. To analyze the chirp echo response, we first extracted the time varying EOD frequency of each fish tested. The fish's EOD was recorded and the instantaneous EOD frequency was computed from the inverse of the timing difference between successive zero crossings. To measure echo responses to small chirp stimuli delivered at different beat phases, we initially habituated the animal to a 4 Hz beat stimulus lasting 60 s in order to minimize the probability of chirp responses being elicited by the beat alone: the chirp rate during a 200 ms window preceding stimulus chirp onset was null (0 ± 0 Hz). We then randomly interspersed chirp stimuli at variable intervals (15 ms ± 3 ms) and the recording was started 200 ms before chirp onset. In total, for each fish, we played 40 chirp stimuli such that each chirp phase was presented five times. Echo response chirps occurring during a 1 s window after stimulus chirp onset were identified as increases in the animal's own EOD frequency that exceeded 30 Hz and were segregated into small (type II) and big (type I) chirps as done previously (*Bastian et al., 2001*). Based on this criterion, almost all echo response chirps were classified as type II (98.7%) while the rest were classified as type I (1.3%). All echo response chirps were included in the analysis. The time of occurrence of echo response chirps was defined as the time at which the EOD frequency excursion was maximal. The echo response chirp rate $CR_{chirp}$ was computed as the number of echo response chirps during a time window of 1 s following the stimulus chirp onset since previous studies have shown that the majority of responses occur during this time window (*Zupanc et al., 2006*). The echo response latency was determined as the timing of the first echo response chirp after chirp stimulus onset. Since the probability of eliciting an echo response chirp was low in general, we averaged the responses to a particular chirp phase obtained from all fish tested to build peri-stimulus time histograms (PSTHs). PSTHs for all single chirp phases were smoothed using a 1 s long boxcar filter. Behavioral responses were quantified as log($CR_{chirp}/CR_{beat}$), where $CR_{beat}$ is the average echo response chirp rate during the beat between 1 and 2 s following stimulus chirp onset, where relatively fewer echo responses tend to occur (*Zupanc et al., 2006*). Invariance scores for behavior were computed as described above for neural responses except that we used the behavioral PSTHs as responses. Error bars were estimated using a sequential bootstrapping method (*Efron, 1979*) with block size 27.

## Statistics

Statistical significance was assessed through one-way analysis of variance (ANOVA) with the Bonferroni method of correcting for multiple comparisons at the p=0.05 level. Values are reported as mean ± SEM.

## Data availability

All data and codes are freely available online (*Metzen et al., 2016*).

## Acknowledgements

We thank Dr. Kathleen E Cullen for critical reading of the manuscript. This research was supported by the Canadian Institutes of Health Research and the Canada Research Chairs (MJC). The funders had no role in study design, data collection and analysis, decision to publish, or preparation of the manuscript.

## Additional information

### Funding

| Funder | Author |
|--------|--------|
| Canadian Institutes of Health Research | Maurice J Chacron |
| Canada Research Chairs | Maurice J Chacron |

The funders had no role in study design, data collection and interpretation, or the decision to submit the work for publication.

### Author contributions

MGM, Conception and design, Acquisition of data, Analysis and interpretation of data, Drafting or revising the article; VH, Acquisition of data, Analysis and interpretation of data, Drafting or revising the article; MJC, Conception and design, Acquisition of data, Drafting or revising the article

### Author ORCIDs

Michael G Metzen, http://orcid.org/0000-0002-2365-4192
Volker Hofmann, http://orcid.org/0000-0002-5149-603X
Maurice J Chacron, http://orcid.org/0000-0002-3032-452X

### Ethics

Animal experimentation: All procedures were approved by McGill University's animal care committee and accordance with guidelines from the Canadian Council on Animal Care under protocol number 5285.

## Additional files

### Major datasets

The following dataset was generated:

| Author(s) | Year | Dataset title | Dataset URL | Database, license, and accessibility information |
|-----------|------|---------------|-------------|--------------------------------------------------|
| Metzen MG, Hofmann V, Chacron MJ | 2015 | Data from: Neural correlations enable invariant coding and perception of natural stimuli | http://dx.doi.org/10.5061/dryad.d8p66 | Available at Dryad Digital Repository under a CC0 Public Domain Dedication. |

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
