## [Decision Letter]

Thank you for submitting your work entitled "Neural correlations enable invariant coding and perception of natural stimuli" for consideration by *eLife*. Your article has been reviewed by three peer reviewers, and the evaluation has been overseen by Ronald L. Calabres as the Reviewing Editor and Eve Marder as the Senior Editor.

The following individual involved in review of your submission has agreed to reveal their identity: Frédéric Theunissen (peer reviewer).

The reviewers have discussed the reviews with one another and the Reviewing Editor has drafted this decision to help you prepare a revised submission.

Summary:

In this report Metzen et al. show for electric fish that neural responses to a communication signal (the chirp) is invariant to the time of occurrence of this signal (phase) relative to the beat envelope produced by the baseline electric field of the two fishes communicating. The results are very convincing: while the response at the level of the afferents is sensitive to the phase, as one moves the neural processing chain responses become more and more invariant to this phase component, potentially explaining how the fish detects chirps and is able to produce the appropriate chirp echo response. It is a beautiful neuroethological study.

Essential revisions:

The data are extensive and appropriately analyzed with relevant statistics. The figures are easy to assimilate and the legends clear. In general the paper is clearly written but there are several concerns about terminology. Materials and methods is extensive but there are concerns about what exactly has been done.

There are major concerns that were discussed by the reviewers and must be addressed before publication in *eLife*.

1) The hypothesized mechanism for how the phase invariance is achieved is not convincing. To quote one of the expert reviewers "The analysis of TS responses is unclear, as are the questions asked and conclusions that follow. Methods: "We limited our recordings to neurons that displayed similar responses to most chirp waveforms." This is a vague criterion for invariance that lacks quantitative rigor. More importantly, to select a subset of invariant neurons and then conclude the neurons were invariant is circular. By contrast, the Figure 4 legend states "Recordings were made from TS neurons that receive balanced excitatory input from both ON and OFF-type ELL hindbrain neurons." (the first paragraph of the subsection “Midbrain TS Neurons are invariant to heterogeneous stimulus waveforms associated with the same electrosensory object” also implies this). Either select neurons based on ON/OFF convergence and then compare the invariance of TS neurons in the convergent/non-convergent groups, or select neurons based on invariance and compare the degree of ON/OFF convergence in the variant/invariant groups. If any relationship is seen, then you can conclude that ON/OFF convergence is associated with invariance. This is a fundamental conclusion (e.g.”Conclusion” section) that is not supported by the data and analyses presented. Further, unless I am missing something, this exact conclusion was already made by Aumentado-Armstrong et al. 2015." In addition, this reviewer stated in consultation "I want to emphasize that it is a significant concern given that one of the major conclusions is that a combination of ON/OFF inputs is what creates the invariance in TS neurons. I do not see any data or analyses that justify this conclusion. It also seems this same conclusion was already reached in their previous publication. Thus, this major conclusion comes across as already discovered and/or an assumption."

2) To quote one of the expert reviewers "[…]the manuscript suffers from over interpretation of results and most importantly does not probe selectivity of neurons to different objects. To expand on the latter point, statements of invariance to identity-preserving stimuli should be supported by measurements of neural responses to both the same and different objects. Here, the authors only study responses to one type of waveform (with different phases). What is missing is the demonstration that while responses of more central neurons exhibit more invariance across different phases for the same type of stimulus waveform they are also exhibit more selectivity to different types of waveforms."

Complementing this concern another expert reviewer wrote "First, the natural variation explored here is along a single dimension, phase. Clearly natural chirps will vary in many other components (amplitude as in the distance and angle between fishes, frequency range of chirp, etc.). Although this is mentioned in the Discussion, it would be nice to get a feel for the actual "natural" variation of these communication signals as produced by two freely swimming and communicating fishes. Also, the correlation might actually not be very sensitive to changes in amplitude? (which would make this a stronger story). Is this something that the authors have measured? Second, I am not convinced that detection of correlated activity is a general principle of invariance in sensory systems. For example, I don't see how it would apply to scale invariance, rotation invariance in visual perception or to intensity invariance or to fundamental frequency invariance in the auditory modality. I am more convinced that the neural signals and mechanisms described here could be general principles to detect signals in noise or to separate two sensory signals (slow and fast as noted in the Discussion). In this sense, the nature of the "invariance" described here appears very specific to the active sense of the electric fish. I still believe that the results are interesting and that general principles of neural computation are revealed but I would be more careful in not overselling the story." This concern can be alleviated by scaling back the claims of the manuscript. In discussion between expert reviewers one wrote "I would support publication if the authors were to rewrite the manuscript around the theme of phase invariance. For that specific form of invariance, perhaps one stimulus waveform is sufficient and their data are adequate. To demonstrate relevance to other sensory systems, the authors could make connection with the known properties of complex cells in the primary visual cortex (which is presently not done)." This seems a useful path to follow in revision.

3) There were some less major concerns about Methods. One reviewer wrote "1) I don't really understand your equation [1]. It seems highly unusual to average a cross-covariance (or correlation) over delay times. You could get values of zero for two functions that have very high coherencies. Also, it sounds like you are actually using the square root of the coherence at DC as a measure of correlation coefficient and not equation [1]. So why mention equation [1]. Please clarify and explain. 2) Doesn't the response, log(A/B) go from -inf to inf? 3) Is the sum in equation [4] over all possible pair-wise comparison? Also would it not make more sense to sum(SMr/SMs)? In this case the differences in pair-wise responses would be "normalized" by the difference in the corresponding stims… (if I understand correctly). Since invariance is such a critical measure for the paper, this entire section on invariance needs to be rewritten to very explicitly state what each component in the invariance metric measures and why this particular form has been chosen.

Another reviewer wrote "The measurement of response was unclear (subsection “Tuning”). Averaging across time and neurons seems to obscure what is a time-varying response. Further, it is not clear why a "response" would be determined by comparing activity within a window to activity within a subwindow. Finally, it is not clear why population-averages were used for comparison, as opposed to pre-stimulus baseline activity. It seems more meaningful and straightforward to ask whether a given neuron increased or decreased its firing rate in response to chirps, which could be determined by comparing each neurons' firing rate during beats with/without a chirp."

There were also some less major concerns about terminology. "Much of the language was unnecessarily opaque, wordy, and jargony:

A) "identity preserving transformation" used throughout. I do not see this as a transformation. It was simply the same stimulus (with a fixed identity not requiring "preservation") being delivered at different times with respect to the beat, resulting in different patterns of sensory input.

B) What is meant by invariant vs. non-invariant, local vs. global invariance? It seems that continuous variation is being lumped into arbitrary categories. Invariance is not categorical (Figure 6 makes this clear).

C) Synchrony vs. correlation. The correlated firing rates of primary afferents is frequently referred to as "synchrony". I view synchrony as coincident firing, not similar changes in firing rates.

D) Discovery of a circuit. In several places, it is stated that a circuit has been revealed/discovered. This circuit was already known. What is being shown here is a particular computation that circuit performs.

E) Electrosensory "objects." There is considerable debate and a lack of consensus over the use of this term. A working definition with relevant citations would help make clear what is meant by the term here.

[Editors' note: further revisions were requested prior to acceptance, as described below.]

Thank you for resubmitting your work entitled "Neural correlations enable invariant coding and perception of natural stimuli" for further consideration at *eLife*. Your revised article has been favorably evaluated by Eve Marder (Senior editor), a Reviewing editor, and two reviewers. The manuscript has been improved but there are some remaining issues that need to be addressed before acceptance, as outlined below:

In the revised version "Neural correlations enable invariant coding and perception of natural stimuli", the authors have addressed some of the comments from the first round and decided than other were beyond the scope of this study. The paper very clearly demonstrates that the "transient" response produced by communication chirp is coded at the level of the afferents with positive (an increase in rate) and negative signals (a decrease in rate) depending on the phase of the chirp relative to the amplitude beat produced by the interfering EOD of the two fishes. That afferent response is therefore phase sensitive. However, the correlated activity of the afferents shows phase invariance (as the rates decrease in synchrony or increase in synchrony). A detector of such correlated activity would be phase invariant. These computations appear to take place in the "next" two levels of neural processing. First at the level of the ELL, ON and OFF cells show increased invariance for all + chirps and -chirps respectively (+ and – chirps are defined according to the phase of the beat). Then at the level of TS, neurons combine information of ON and OFF cells to generate positive responses to most chirps irrespective of phase. Thus phase invariance is achieved simply by first a weighted sum of afferent responses into two groups and then a sum of these two responses. This computation appears to be similar (both in spirit and connectivity) to the "hierarchical models" that have been proposed for describing complex cells in V1 (see Martinez and Alonso, 2003). It may not be exactly a "correlator" detector or a more general neural computation for other forms of invariance but this is now appropriately discussed.

There remains a major concern that was discussed by the reviewers and must be addressed before publication in *eLife*. It is not clear how their model/conclusions fit with the data in Figure 3 and Figure 4. It is easy to imagine how appropriate weighting of inputs from one stage of processing could give rise to the increased invariance observed at the next stage, and this is an important finding. However, the way in which this happens does not seem as simple as the narrative presented: (31) ELL ON-cells being invariant to +chirps but not -chirps; (66) ELL OFF-cells being invariant to -chirps but not +chirps; and (6) TS cells integrating ON- and OFF-cell inputs to establish global chirp invariance. The data in Figure 3 and Figure 4 and the accompanying descriptions in the text do not show this convincingly. The authors either need to show that the +chirp and -chirp differences in Figure 3 are significant (perhaps with a 2-way ANOVA), or present their conclusions in such a way that they do not depend on their being a difference here. Similarly, they need to show that the +chirp and -chirp differences in Figure 4 are not significant, or present their conclusions in such a way that they do not depend on their being no difference here.

More minor is the suggestion from the last review about showing more natural variability. This would seem to make the point on perception more relevant rather than being outside the scope of the report.

---

## [Author Response]

Essential revisions:

*The data are extensive and appropriately analyzed with relevant statistics. The figures are easy to assimilate and the legends clear. In general the paper is clearly written but there are several concerns about terminology. Materials and methods is extensive but there are concerns about what exactly has been done. There are major concerns that were discussed by the reviewers and must be addressed before publication in eLife. 1) The hypothesized mechanism for how the phase invariance is achieved is not convincing. To quote one of the expert reviewers "The analysis of TS responses is unclear, as are the questions asked and conclusions that follow. Methods: "We limited our recordings to neurons that displayed similar responses to most chirp waveforms." This is a vague criterion for invariance that lacks quantitative rigor. More importantly, to select a subset of invariant neurons and then conclude the neurons were invariant is circular. By contrast, the Figure 4 legend states "Recordings were made from TS neurons that receive balanced excitatory input from both ON and OFF-type ELL hindbrain neurons." (the first paragraph of the subsection “Midbrain TS Neurons are invariant to heterogeneous stimulus waveforms associated with the same electrosensory object” also implies this). Either select neurons based on ON/OFF convergence and then compare the invariance of TS neurons in the convergent/non-convergent groups, or select neurons based on invariance and compare the degree of ON/OFF convergence in the variant/invariant groups. If any relationship is seen, then you can conclude that ON/OFF convergence is associated with invariance. This is a fundamental conclusion (e.g.”Conclusion” section) that is not supported by the data and analyses presented. Further, unless I am missing something, this exact conclusion was already made by Aumentado-Armstrong et al.2015." In addition, this reviewer stated in consultation "I want to emphasize that it is a significant concern given that one of the major conclusions is that a combination of ON/OFF inputs is what creates the invariance in TS neurons. I do not see any data or analyses that justify this conclusion. It also seems this same conclusion was already reached in their previous publication. Thus, this major conclusion comes across as already discovered and/or an assumption."*

We understand and agree with these concerns. To address them, we now compute phase invariance across all recorded TS neurons. On average, phase invariance scores were significantly larger in TS than in ELL. This was because, while some TS neurons displayed phase invariance scores comparable to those of ELL neurons, others actually displayed larger values (max ELL: 0.69, min ELL: 0.11; max TS: 0.84, min TS: 0.37). We now mention in the Discussion that a previous study (Aumentado-Armstrong et al. 2015) has shown that the later subset of neurons, (i.e. those displaying the greatest phase invariance scores), receive balanced input from ON and OFF type ELL pyramidal cells (subsection “Coding of natural electro-communication stimuli”, third paragraph).

*2) To quote one of the expert reviewers "…the manuscript suffers from over interpretation of results and most importantly does not probe selectivity of neurons to different objects. To expand on the latter point, statements of invariance to identity-preserving stimuli should be supported by measurements of neural responses to both the same and different objects. Here, the authors only study responses to one type of waveform (with different phases). What is missing is the demonstration that while responses of more central neurons exhibit more invariance across different phases for the same type of stimulus waveform they are also exhibit more selectivity to different types of waveforms." Complementing this concern another expert reviewer wrote "First, the natural variation explored here is along a single dimension, phase. Clearly natural chirps will vary in many other components (amplitude as in the distance and angle between fishes, frequency range of chirp, etc.). Although this is mentioned in the Discussion, it would be nice to get a feel for the actual "natural" variation of these communication signals as produced by two freely swimming and communicating fishes. Also, the correlation might actually not be very sensitive to changes in amplitude? (which would make this a stronger story). Is this something that the authors have measured? Second, I am not convinced that detection of correlated activity is a general principle of invariance in sensory systems. For example, I don't see how it would apply to scale invariance, rotation invariance in visual perception or to intensity invariance or to fundamental frequency invariance in the auditory modality. I am more convinced that the neural signals and mechanisms described here could be general principles to detect signals in noise or to separate two sensory signals (slow and fast as noted in the Discussion). In this sense, the nature of the "invariance" described here appears very specific to the active sense of the electric fish. I still believe that the results are interesting and that general principles of neural computation are revealed but I would be more careful in not overselling the story." This concern can be alleviated by scaling back the claims of the manuscript. In discussion between expert reviewers one wrote "I would support publication if the authors were to rewrite the manuscript around the theme of phase invariance. For that specific form of invariance, perhaps one stimulus waveform is sufficient and their data are adequate. To demonstrate relevance to other sensory systems, the authors could make connection with the known properties of complex cells in the primary visual cortex (which is presently not done)." This seems a useful path to follow in revision.*

We understand this important concern and have rewritten the manuscript around phase invariance as suggested. We appreciate the interest in understanding how electrosensory neurons respond to other chirp attributes such as duration and frequency increase and now more clearly mention that these fall outside the scope of our paper and thus should be the focus of future studies. We now also connect the results with complex cells of the primary visual cortex as suggested and focus on their phase invariance properties (subsection “Implications for other systems”, first paragraph).

*3) There were some less major concerns about Methods. One reviewer wrote "1) I don't really understand your equation [1]. It seems highly unusual to average a cross-covariance (or correlation) over delay times. You could get values of zero for two functions that have very high coherencies. Also, it sounds like you are actually using the square root of the coherence at DC as a measure of correlation coefficient and not equation [1]. So why mention equation [1]. Please clarify and explain. 2) Doesn't the response, log(A/B) go from -inf to inf? 3) Is the sum in equation [4] over all possible pair-wise comparison? Also would it not make more sense to sum(SMr/SMs)? In this case the differences in pair-wise responses would be "normalized" by the difference in the corresponding stims… (if I understand correctly). Since invariance is such a critical measure for the paper, this entire section on invariance needs to be rewritten to very explicitly state what each component in the invariance metric measures and why this particular form has been chosen.*

We understand the concern and now mention that we indeed use equation (66) to compute the cross-correlation coefficient (Methods). However, this definition is actually equivalent to equation (31), which was used in previous studies (e.g. Shadlen and Newsome 1998). We now mention this in the Methods and have deleted equation (31) to avoid further confusion.

As suggested, we have recomputed the phase invariance score by normalizing the distance between responses by the distance between the corresponding stimulus waveforms for all pairings and have updated figures accordingly. This did not affect our results qualitatively. The Methods section now includes more details as to how phase invariance scores were computed.

Another reviewer wrote "The measurement of response was unclear (subsection “Tuning”). Averaging across time and neurons seems to obscure what is a time-varying response. Further, it is not clear why a "response" would be determined by comparing activity within a window to activity within a subwindow. Finally, it is not clear why population-averages were used for comparison, as opposed to pre-stimulus baseline activity. It seems more meaningful and straightforward to ask whether a given neuron increased or decreased its firing rate in response to chirps, which could be determined by comparing each neurons' firing rate during beats with/without a chirp."

We understand these concerns and, as suggested, now compute responses by comparing the firing rate during a small time window after chirp onset to the average firing rate during the beat. Further, as suggested, we now compute all measures (i.e. both response and phase invariance score) for individual neurons and then average the results over the population. The Methods and figures have been updated accordingly. Overall, these changes did not affect our results qualitatively.

There were also some less major concerns about terminology. "Much of the language was unnecessarily opaque, wordy, and jargony: A) "identity preserving transformation" used throughout. I do not see this as a transformation. It was simply the same stimulus (with a fixed identity not requiring "preservation") being delivered at different times with respect to the beat, resulting in different patterns of sensory input.

We have rewritten the paper around the theme of phase invariance, as suggested above, and have thus removed all occurrences of the term “identity preserving transformation” in the text.

*B) What is meant by invariant vs. non-invariant, local vs. global invariance? It seems that continuous variation is being lumped into arbitrary categories. Invariance is not categorical (Figure 6 makes this clear).*

We agree and have rewritten the paper accordingly. In particular, we now focus on the degree of phase invariance as quantified by our measure.

*C) Synchrony vs. correlation. The correlated firing rates of primary afferents is frequently referred to as "synchrony". I view synchrony as coincident firing, not similar changes in firing rates.*

We understand the concern and emphasize that the correlation coefficient was computed on the spike trains and not on the time dependent firing rates in both the Methods (subsection “Correlation between the spiking activities of electrosensory afferents”) and Results (subsection “Correlated activity between receptor afferents is phase invariant”). We have also replaced “synchrony” by “similarity” in the aforementioned subsection to avoid further confusion.

D) Discovery of a circuit. In several places, it is stated that a circuit has been revealed/discovered. This circuit was already known. What is being shown here is a particular computation that circuit performs.

This has been corrected and we now mention that we are uncovering computations performed by electrosensory neurons in order to give rise to phase invariance.

*E) Electrosensory "objects." There is considerable debate and a lack of consensus over the use of this term. A working definition with relevant citations would help make clear what is meant by the term here.*

We have removed all mentions of “electrosensory objects” throughout the manuscript and now instead refer to chirps or natural communication stimuli throughout.

[Editors' note: further revisions were requested prior to acceptance, as described below.]

The manuscript has been improved but there are some remaining issues that need to be addressed before acceptance, as outlined below: In the revised version "Neural correlations enable invariant coding and perception of natural stimuli", the authors have addressed some of the comments from the first round and decided than other were beyond the scope of this study. The paper very clearly demonstrates that the "transient" response produced by communication chirp is coded at the level of the afferents with positive (an increase in rate) and negative signals (a decrease in rate) depending on the phase of the chirp relative to the amplitude beat produced by the interfering EOD of the two fishes. That afferent response is therefore phase sensitive. However, the correlated activity of the afferents shows phase invariance (as the rates decrease in synchrony or increase in synchrony). A detector of such correlated activity would be phase invariant. These computations appear to take place in the "next" two levels of neural processing. First at the level of the ELL, ON and OFF cells show increased invariance for all + chirps and -chirps respectively (+ and – chirps are defined according to the phase of the beat). Then at the level of TS, neurons combine information of ON and OFF cells to generate positive responses to most chirps irrespective of phase. Thus phase invariance is achieved simply by first a weighted sum of afferent responses into two groups and then a sum of these two responses. This computation appears to be similar (both in spirit and connectivity) to the "hierarchical models" that have been proposed for describing complex cells in V1 (see Martinez and Alonso, 2003). It may not be exactly a "correlator" detector or a more general neural computation for other forms of invariance but this is now appropriately discussed.

*There remains a major concern that was discussed by the reviewers and must be addressed before publication in eLife. It is not clear how their model/conclusions fit with the data in Figure 3 and Figure 4. It is easy to imagine how appropriate weighting of inputs from one stage of processing could give rise to the increased invariance observed at the next stage, and this is an important finding. However, the way in which this happens does not seem as simple as the narrative presented: (31) ELL ON-cells being invariant to +chirps but not -chirps; (66) ELL OFF-cells being invariant to - chirps but not + chirps; and (6) TS cells integrating ON- and OFF-cell inputs to establish global chirp invariance. The data in Figure 3 and Figure 4 and the accompanying descriptions in the text do not show this convincingly. The authors either need to show that the +chirp and -chirp differences in Figure 3 are significant (perhaps with a 2-way ANOVA), or present their conclusions in such a way that they do not depend on their being a difference here. Similarly, they need to show that the +chirp and -chirp differences in Figure 4 are not significant, or present their conclusions in such a way that they do not depend on their being no difference here.*

We understand this major concern and have now performed the additional required analyses to show that ELL ON and OFF-cells are more invariant to same type chirps than to opposite type chirps. Intuitively, this makes sense as ON (OFF)-type cells respond similarly and with excitation (inhibition) to “+chirps” and similarly but with inhibition (excitation) to “-chirps”. Moreover, we now show that, for TS neurons, invariance scores computed from same type chirps and opposite type chirps are not significantly different from one another. We have updated the main text as well as Figure 2, Figure 3, Figure 4.

More minor is the suggestion from the last review about showing more natural variability. This would seem to make the point on perception more relevant rather than being outside the scope of the report.

We now include example actual chirp waveforms recorded from behaving fish (Figure 1—figure supplement 1).